# Asymmetric formal sp²-hydrocarbonations of dienes and alkynes via palladium hydride catalysis

Ming-Qiao Tang[1,3], Zi-Jiang Yang[1,3] & Zhi-Tao He [1,2] ✉

Transition metal-catalyzed asymmetric hydrofunctionalizations of unsaturated bonds via π-η³ substitution have emerged as a reliable method to construct stereogenic centers, and mainly rely on the use of heteroatom-based or carbon nucleophiles bearing acidic C-H bonds. In comparison, sp² carbon nucleophiles are generally not under consideration because of enormous challenges in cleaving corresponding inert sp² C-H bonds. Here, we report a protocol to achieve asymmetric formal sp² hydrocarbonations, including hydroalkenylation, hydroallenylation and hydroketenimination of both 1,3-dienes and alkynes via hydroalkylation and Wittig reaction cascade. A series of unachievable motifs via hydrofunctionalizations, such as di-, tri- and tetra-substituted alkenes, di-, tri- and tetra-substituted allenes, and tri-substituted ketenimines in allyl skeletons are all facilely constructed in high regio-, dia-stereo- and enantioselectivities with this cascade design. Stereodivergent synthesis of all four stereoisomers of 1,4-diene bearing a stereocenter and Z/E-controllable olefin unit highlights the power of present protocol. An interesting mechanistic feature is revealed that alkyne actually undergoes hydrocarbonation via the formation of conjugated diene intermediate, different from conventional viewpoint that the hydrofunctionalization of alkynes only involves allene species.

Transition metal-catalyzed asymmetric hydrofunctionalizations of conjugated dienes via π-η³ substitution have emerged as a reliable strategy to construct allylic stereogenic centers, due to their 100% atom-economy, good stereocontrol performance and easily accessible substrates[1–5]. Different types of nucleophiles have thus been successfully introduced to the allylic site to prepare C – C, C – N, C – S, C – P and C – O bonds etc. stereoselectively[6–33]. Generally, substrate containing a slightly acidic C-H or heteroatom-H bond is required for the facile cleavage of the hydrogen atom to generate corresponding nucleophilic center and then to react with η³-allyl metal intermediate via outer-sphere or inner-sphere substitution[1–5]. Therefore, substrates such as olefins, allenes, and ketenimines bearing inert sp² C – H bonds

are usually not considered as suitable nucleophiles for asymmetric hydrofunctionalizations of 1,3-dienes. Similarly, such processes have not been reported for alkynes as another abundant unsaturated hydrocarbons.

Different from allylic substitution pathway aforementioned, an elegant strategy involving cyclometallation and β-H elimination process has been developed to realize Ni- or Co-catalyzed asymmetric hydrovinylation of 1,3-dienes (Fig. 1a)[34–45]. The scope was mainly limited to the use of ethylene as the only olefin source. Until 2017, RajanBabu extended to the adoption of acrylates for enantioselective Co-catalyzed hydroalkenylation of 1,3-dienes[42]. Later, Ho realized Ni-catalyzed related transformation with mono-substituted olefins as the

[1]CAS Key Laboratory of Synthetic Chemistry of Natural Substances, Shanghai Institute of Organic Chemistry, University of Chinese Academy of Sciences, Shanghai 200032, China. [2]School of Chemistry and Materials Science, Hangzhou Institute for Advanced Study, University of Chinese Academy of Sciences, Hangzhou 310024, China. [3]These authors contributed equally: Ming-Qiao Tang, Zi-Jiang Yang. ✉e-mail: hezt@sioc.ac.cn

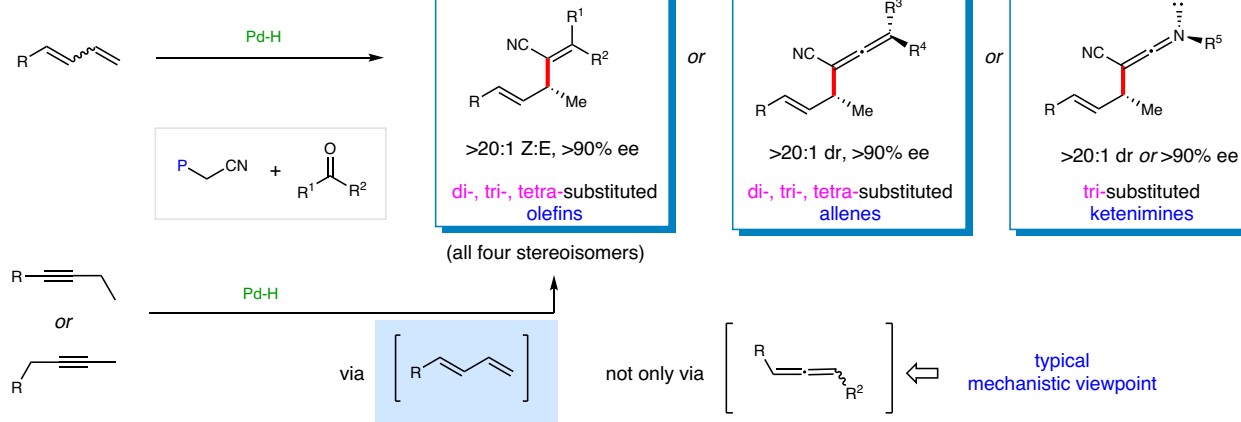

**Fig. 1 | Asymmetric hydroalkenylation of unsaturated bonds and our strategy. a** Ni- or Co-catalyzed asymmetric hydrovinylation of 1,3-dienes. **b** Our strategy for the introduction of di-, tri-, tetra-substituted olefins, allenes & ketenimines via cascade hydrofunctionalization/Wittig reaction. ee enantiomeric excess.

coupling partners[44]. However, the introduction of diverse olefins, especially highly functionalized alkene units, is difficult. The major challenges blocking the use of multi-substituted alkenes as coupling partners might arise from the increased steric hindrance leading to low reactivity and complex regiocontrol from metal hydride-catalyzed isomerization of both unsaturated substrates. Thus, a feasible protocol to introduce olefins with multiple substituents to conjugated dienes, and also to other unsaturated bonds such as alkynes, via asymmetric hydrofunctionalization pathway is still absent.

On the other hand, allene is a versatile building block in organic synthesis and a skeleton abundant in natural products[46–48]. The construction of enantioenriched 1,4-enallenes should be valuable and intriguing, considering the enormous downstream transformation potential of these two differentiated unsaturated bonds. The hydroallenylation of unsaturated bonds might provide a straightforward route to prepare such motifs but remains unreported. Similarly, stereoselective hydroketenimination to synthesize allyl ketenimines as another type of valuable skeleton is also unknown[49–51].

As Pd-catalyzed asymmetric hydrofunctionalizations of unsaturated hydrocarbons[1,52–67] have repeatedly been demonstrated efficient for C − C bond formation, we envisioned that an ylide-involved hydrofunctionalization as the precursor of allyl alkenes[68,69], allenes and ketenimines might provide a general route for challenging asymmetric sp²-hydrocarbonations (Fig. 1b). However, the identification of a suitable ylide for hydrofunctionalization and the perfect merger of the cascade process remain uncertain. In addition, the construction of multi-substituted alkenes, allenes, and ketenimines in high stereocontrol is also very challenging[70].

Herein, we describe a reliable strategy to achieve these challenging formal sp²-hydrocarbonations via hydroalkylation and Wittig reaction cascade. Previously unachievable motifs via hydrofunctionalization, including di-, tri- and tetra-substituted alkenes, di-, tri- and tetra-substituted allenes and tri-substituted ketenimines are all smoothly introduced to conjugated dienes and alkynes in high stereocontrol. Stereodivergent synthesis of all four isomers of skipped dienes bearing a stereocenter and configuration-controllable olefin unit highlights the power of present method. Mechanistic studies

reveal that alkyne might undergo the hydrofunctionalization via the formation of 1,3-diene intermediate, not only involving allene intermediate.

## Results

### Reaction development for formal hydroalkenylation

We initiated the formal hydroalkenylation with diene **1a** as the electrophile, phosphonate **2** combined with aqueous formaldehyde **3a** as alkene surrogate under palladium catalysis (Table 1). Based on our previous work on Pd-catalyzed asymmetric hydrofunctionalizations[19–21,60,61,67], a series of JosiPhos-type chiral ligands were evaluated first (entries 1 − 4). The alkenylation product **4a** was smoothly generated with **L9 − L11**, among which a moderate yield and high enantioselectivity for **4a** was observed with **L11** (entry 4). Other types of chiral ligands that performed well in previous asymmetric hydrofunctionalizations[7,20] showed no reactivity (entries 5−7). The screening of diverse solvents led to the erosion of reaction efficiency (entries 8−11). As catalytic amount of acid was reported to be favorable for the generation of PdH catalyst[8,57,63], 20% diphenylphosphinic acid as a co-catalyst was used. Indeed, **4a** was provided in a higher 68% yield and 95% ee (entry 12). Finally, both elongated reaction time and elevated concentration facilitated the transformation and furnished **4a** with optimal yield and stereoselectivity (entry 13). In addition, other types of phosphonate nucleophiles such as **2a** and **2b** bearing a ketone or ester unit instead of a smaller and more electron-deficient cyano group were not effective for this reaction, presumably due to the unsuitable p$K_a$ and increased steric hindrance[60].

### Scope for asymmetric formal hydroalkenylation

With the optimized conditions in hand, the scope for one-pot asymmetric hydroalkenylation of dienes were first evaluated and the results are summarized in Fig. 2. As the geometry of internal olefin in diene substrate **1** did not affect the reaction, presumably due to the facile isomerization of (Z)-**1** into (E)-**1** (see Supplementary Information 8.6 for details), Z/E mixtures of **1** were directly used as the substrates for present transformation. A variety of dienes containing functional groups featuring different steric hindrance and electronic characters

**Table 1 | Reaction development for formal hydroalkenylation of 1,3-diene**

| Entry | Ligand | Solvent | Additive | Yield (%)[a] | ee (%)[b] |
|---|---|---|---|---|---|
| 1 | L1-8 | DCM | no | trace | |
| 2 | L9 | DCM | no | 54 | 84 |
| 3 | L10 | DCM | no | 46 | 86 |
| 4 | L11 | DCM | no | 50 | 94 |
| 5 | L12 | DCM | no | trace | |
| 6 | L13 | DCM | no | trace | |
| 7 | L14 | DCM | no | trace | |
| 8 | L11 | MeCN | no | 24 | 94 |
| 9 | L11 | THF | no | n.d. | |
| 10 | L11 | PhCF₃ | no | 36 | 94 |
| 11 | L11 | hexane | no | 24 | 94 |
| 12 | L11 | DCM | Ph₂P(O)OH | 68 | 95 |
| 13[c,d] | L11 | DCM | Ph₂P(O)OH | 82 | 94 |
| 14[c,e] | L11 | DCM | Ph₂P(O)OH | trace | |
| 15[c,f] | L11 | DCM | Ph₂P(O)OH | n.d. | |

*Cy* cyclohexyl, *n.d.* not detected, *DCM* dichloromethane, *THF* tetrahydrofuran, *RT* room temperature

[a]The reaction was carried out with HCHO (37% aqueous solution, 12 equiv) for the second step. Yield was determined by ¹H NMR with 1,3,5-trimethoxybenzene as the internal standard.

[b]Determined by HPLC analysis.

[c]2(2.0 equiv) and DCM (2 M) for 48 h were adopted for first step. Additional Et₃N (1.0 equiv) was used for the second step.

[d]Isolated yield.

[e]2a was used instead of 2.

[f]Nucleophile 2b was used instead of 2. NaBArF₄, ArF = 3,5-(CF₃)₂Ph.

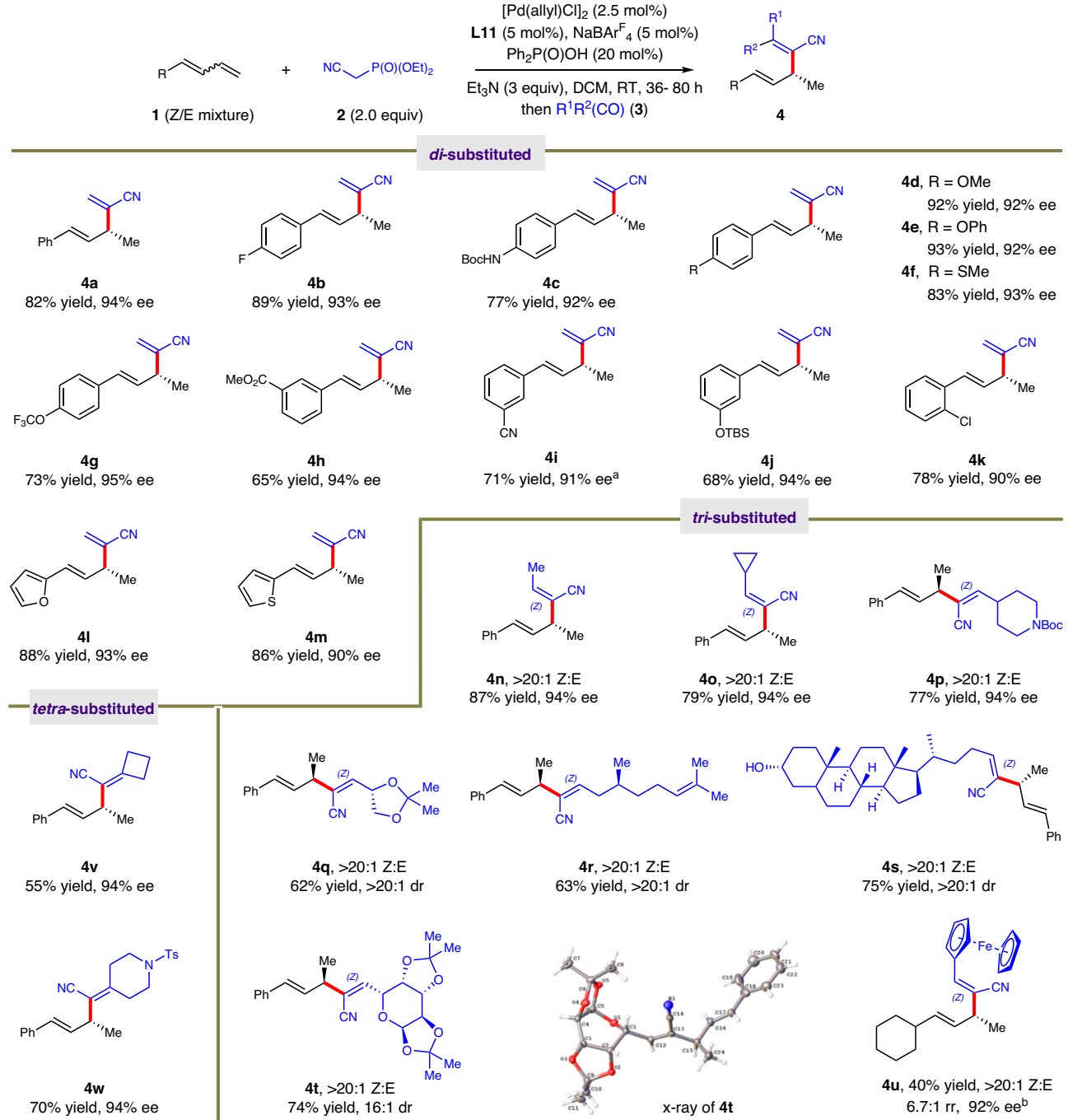

**Fig. 2 | Scope for asymmetric formal hydroalkenylation of 1,3-dienes.** Isolated yield for 0.1 mmol scale reaction. The ee values were determined by HPLC analysis. NaBAr$^F_4$, Ar$^F$ = 3,5-(CF$_3$)$_2$Ph. For di-substituted olefin construction: HCHO and Et$_3$N were used at RT. For tri- & tetra-substituted olefin construction: R$^1$R$^2$(CO) & $^i$PrMgCl were used at RT. See the supporting information for details. $^a$At 50 °C. $^b$[Pd(allyl)Cl]$_2$ (5 mol%), **L11** (10 mol%) and NaBAr$^F_4$ (10 mol%) were used.

showed high compatibility with the established protocol. For example, the aryl groups in diene substrates with halide, protected amine, ether, thioether, OCF$_3$, ester, cyano, OTBS as substituents underwent the one-pot hydroalkenylation in 65–93% yield and 90–95% ee (**4a** – **4k**). Heteroaryl-derived conjugated diene also displayed good tolerance, providing allyl products with *di*-substituted olefins in >80% yield and >90% ee (**4 l, 4 m**).

When different aldehydes were used, the scope for the construction of tri-substituted alkenes was checked. In this context, a stronger base $^i$PrMgCl was required instead of Et$_3$N to promote the alkene formation. Then Z-form olefins in the enantioenriched allyl

products were prepared smoothly via the transformation in high stereoselectivity (**4n** – **4 u**). A series of units including alkyl, cyclopropyl, piperidine and glyceraldehyde derivative on the prepared tri-substituted alkenes had no discernible influence on the formal hydroalkenylations, delivering corresponding products **4n** – **4q** in 62–87% yield, 94% ee as single Z-form geometric isomer. When complex natural products such as lithocholic acid, citronellal and sugar derivatives were used as alkene sources, tri-substituted alkene skeletons were achieved in high efficiency and stereocontrol (**4r** – **4t**). The absolute configuration of **4t** was revealed by X-ray crystal analysis, which also elucidated the configuration of Z-olefin. In addition, when

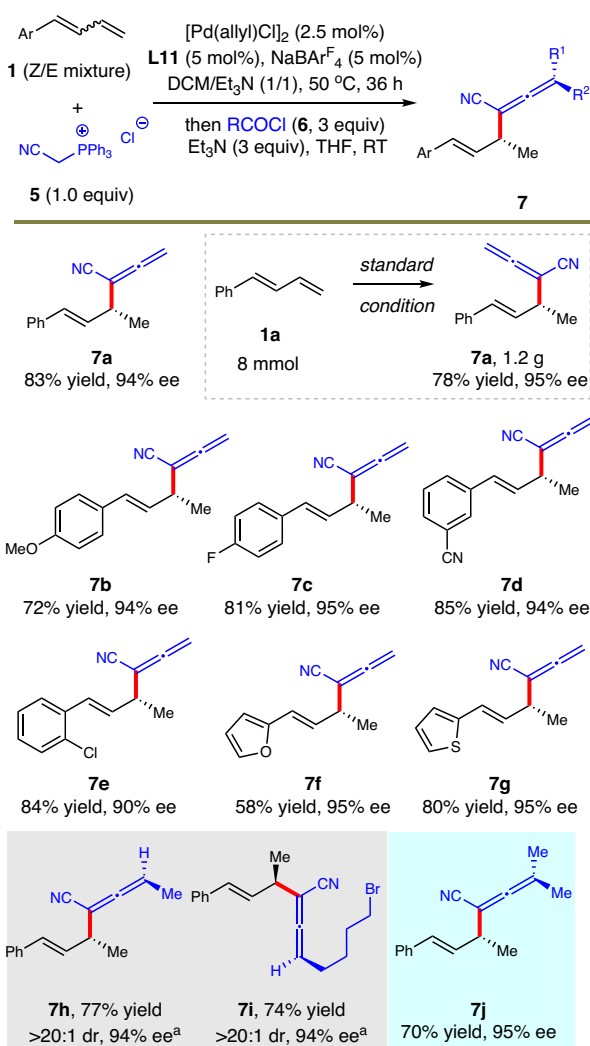

**Fig. 3 | Scope for asymmetric formal hydroallenylation of 1,3-dienes.** Isolated yield. The ee was determined by HPLC analysis. [a]The absolute configuration of the axis in the product was not determined.

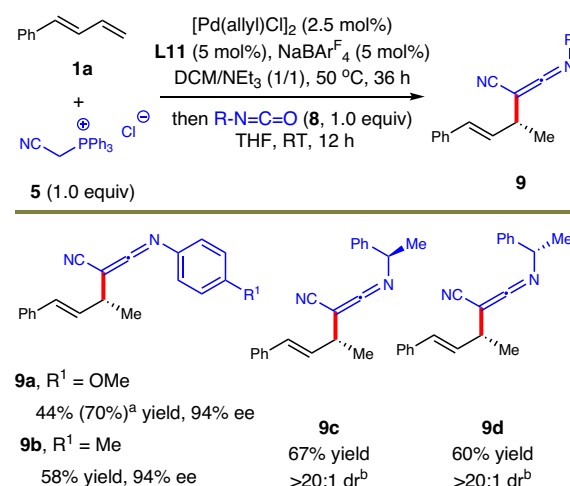

**Fig. 4 | Scope for asymmetric formal hydroketenimination of 1,3-dienes.** Isolated yield. The ee was determined by HPLC analysis. [a]Determined by [1]H NMR. [b]**8** (2.0 equiv) was used.

Next, the scope and robustness of aforementioned hydroallenylation were checked and the results are summarized in Fig. 3. A gram-scale test was carried out with 8 mmol of **1a**, and 1.2 gram of allene product **7a** was easily synthesized in 78% yield and 95% ee, highlighting the reliability of this method. Then, several aryl dienes bearing ether, halide and cyano unit were selected to undergo the hydrofunctionalizations. All corresponding products were afforded in 72–85% yield and 90–95% ee (**7b–7e**). Moreover, the protocol also tolerated heteroaryl-derived dienes well (**7f–7g**). As the coupling partner ketene for Wittig reaction was generated in situ from acyl chloride under Et₃N, we assumed that monosubstituted ketenes might lead to the formation of tri-substituted allenes. However, whether the pre-existing carbon stereocenter could efficiently control the formed vicinal axial chirality of allene unit with no catalyst involved was uncertain. Fortunately, allyl products with tri-substituted allene moieties were obtained in good yields and excellent stereocontrol (**7h, 7i**). Even for the preparation of tetra-substituted allene compound **7j**, present route showed high reactivity and stereoselectivity.

## Asymmetric formal hydroketenimination
Ketenimine is a unique structure as isoelectronic species of ketene and allene[49–51]. It is often not very stable and widely works as synthetic intermediate for diverse downstream transformations[49–51,71,72]. Inspired by the smooth construction of chiral allenes via hydroallenylation protocol, we assumed that similar pathway should also be suitable for the generation of enantioenriched ketenimine skeletons. Indeed, under the standard conditions established for allene formation but with isocyanate instead of acyl chloride as the substrate, fully substituted ketenimines were produced stereoselectively (Fig. 4). For example, allyl ketenimines **9a** and **9b** were prepared via this one-pot hydrofunctionalization process in moderate yields and excellent enantioselectivities. When isocyanates bearing a vicinal stereocenter were employed as the source of ketenimines, similar high diastereoselectivities were observed for products **9c** and **9d** without any influence of matched/mismatched interaction between the corresponding isocyanates and Wittig reagent intermediate. This design represents a reliable route to construct scarcely studied chiral ketenimine skeletons.

## Asymmetric hydroalkenylation of alkynes
Transition metal-catalyzed asymmetric hydrofunctionalizations of alkynes have been reasonably studied[1]. However, the construction of enantioenriched skipped dienes via this pathway, that is,

this hydroalkenylation process was further extended to more challenging transformations, i.e., the introduction of tetra-substituted olefins to allylic sites enantioselectively, corresponding scaffolds **4 v** and **4w** were furnished smoothly in 55–70% yield and 94% ee. It should be noted that alkyl substituted 1,3-diene as the substrate was also suitable, giving **4 u** in an eroded yield and reasonable stereoselectivity. Internal dienes did not work for the present reaction.

## Asymmetric formal hydroallenylation
Compared with alkene, allene unit is usually much more reactive towards transition metal catalyst. In addition, the control of axial chirality further increased the challenge in the introduction of an allene group. Considering the great application potential of enantioenriched allene skeletons and the absence of hydroallenylation reaction, we moved on to explore the possibility of Pd-catalyzed formal hydroallenylation reaction of 1,3-dienes (see Supplementary Information 3.3 for details). Under the standard conditions established for hydroalkenylation, only trace amount of product **7a** was observed. We assumed that electron-deficient product **7a** might not be compatible with the strong basic condition for the second Wittig reaction. Then Wittig reagent **5** was chosen and weak base Et₃N was found to be strong enough to promote the transformation. After a series of condition optimizations, we finally prepared **7a** in 83% yield and 94% ee.

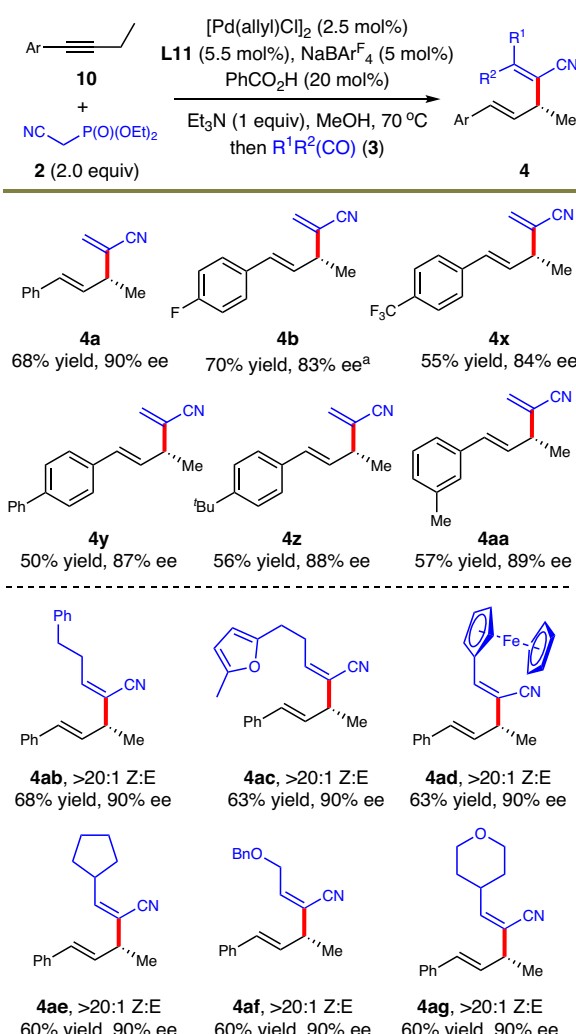

**Fig. 5 | Asymmetric formal hydroalkenylation of alkynes.** Isolated yield. The ee was determined by HPLC analysis. [a]The concentration of hydroalkylation was 2 M without NaBAr$^F_4$.

hydroalkenylation of alkynes, has never been reported. We thus continued our attempts to solve this long-standing challenge. With slight adjustment of the original standard conditions for hydroalkenylation of 1,3-dienes, we realized this one-pot hydroalkenylation process of alkynes (Fig. 5). Various aryl alkynes containing electron-rich or electron-deficient functional groups underwent the hydroalkenylation smoothly and furnished 1,4-diene products in moderate yields and good enantioselectivities (**4a–4aa**). When different aldehydes were utilized, a set of tri-substituted alkene motifs were constructed at allylic positions in reasonable yields and high enantioselectivities as exclusively Z-form isomers (**4ab–4ag**).

### Mechanistic studies

A general mechanism for transition metal-catalyzed asymmetric hydrofunctionalizations of alkynes with outer-sphere nucleophiles involves the sequential conversion of alkyne into allene intermediate, hydrometallation of allene to generate thermodynamically stable η$^3$-allylmetal species and final allylic substitution[1,73]. Following the seminal work by Yamamoto[74–76], progress has been made on this area under palladium catalysis based on aforementioned mechanistic proposal. However, when we analyzed the results of hydroallenylation reaction of alkyne **10a**, trace amount of conjugated diene **1a** was observed (Fig. 6A). With allene intermediate **10a'** as the substrate for

hydrofunctionalization, diene **1a** was obtained in 18% yield. These facts led us to hypothesize that diene might be a critical intermediate in the transformation of alkyne. Next, another alkyne **11** was employed as the substrate (Fig. 6B). Indeed, internal diene **12** which could not undergo the hydroalkenylation was then provided in 46% yield, and no allene species was detected. To further corroborate the hypothesis aforementioned, 1,3-diene **1a** was directly adopted as the substrate under standard condition established for alkyne hydroallenylation (Fig. 6C). Undoubtedly, corresponding product **4a** was provided in 73% yield and 90% ee, comparable to the results observed for **4a** in Fig. 5. Finally, when deuterated **2a** was utilized as the nucleophile in different solvents, obvious deuteration was detected at the terminal methyl group of **4a** for both cases, further showing the generation of diene intermediate via possible β-H elimination process (Fig. 6D). This fact also indicated that the disassociation of PdH catalyst from diene intermediate should be facile, so that PdD could re-coordinate with diene and finish the introduction of D atom to the terminal methyl site.

Based on these experimental results, we proposed a possible mechanism (Fig. 6E). Alkyne **10a** was first converted into allene (**int-2**), which quickly underwent irreversible hydrometallation to generate thermodynamically stable η$^3$-Pd species **int-3**. As the final substitution might be relatively slow, the **int-3** underwent a quick balance with 1,3-diene intermediate **int-4**. Finally, allylic substitution finished this catalytic cycle. This uncovered mechanistic character should be valuable to elucidate other related processes involved hydrofunctionalizations of alkynes.

Inspired by this unclassical mechanistic route, we assumed that an internal alkyne **13** might also undergo the generation of 1,3-diene intermediate to work as a suitable substrate for present methodology[77–79]. We did prepare **4a** from **13** in a moderate yield and high enantioselectivity (Fig. 6F). However, when terminal alkyl alkyne **14** was adopted as the electrophile, no product was detected. These facts suggested that the existence of vicinal aryl unit to alkyne is important, due to the easy formation of aryl-conjugated diene intermediate.

### Transformations

Considering the same intermediate was shared, a convergent transformation involving Z/E mixtures of **1a, 10a** and **13** was carried out. The product **4a** was isolated in 60% yield and 90% ee, highlighting the broad reaction compatibility (Fig. 7A). In addition, as the allene unit in product **7a** could be stereoselectively reduced to generate E-form olefin, stereodivergent synthesis of all four stereoisomers of **4n** from **1a** was conducted. All these isomers were prepared in high stereocontrol (Fig. 7B).

### Discussion

We have established a general strategy for asymmetric formal sp$^2$-hydrocarbonations of both conjugated dienes and alkynes, including hydroalkenylations, hydroallenylations and hydroketeniminations. A series of challenging allyl compounds via hydrofunctionalization, such as di-, tri- and tetra-substituted alkenes, di-, tri- and tetra-substituted allenes and tri-substituted ketenimines in allyl skeletons are smoothly constructed in high yields and generally >20:1 Z/E, >20:1 dr and >90% ee. Stereodivergent synthesis of all four stereoisomers of an allyl motif bearing a stereocenter and geometry-controllable olefin showcases the value of present method. Mechanistic experiments reveal that the stereoselective hydrofunctionalization of alkynes actually undergoes the formation of conjugated diene species, different from the typical viewpoint about alkyne hydrofunctionalization, which is proposed to only involve the formation of allene intermediate.

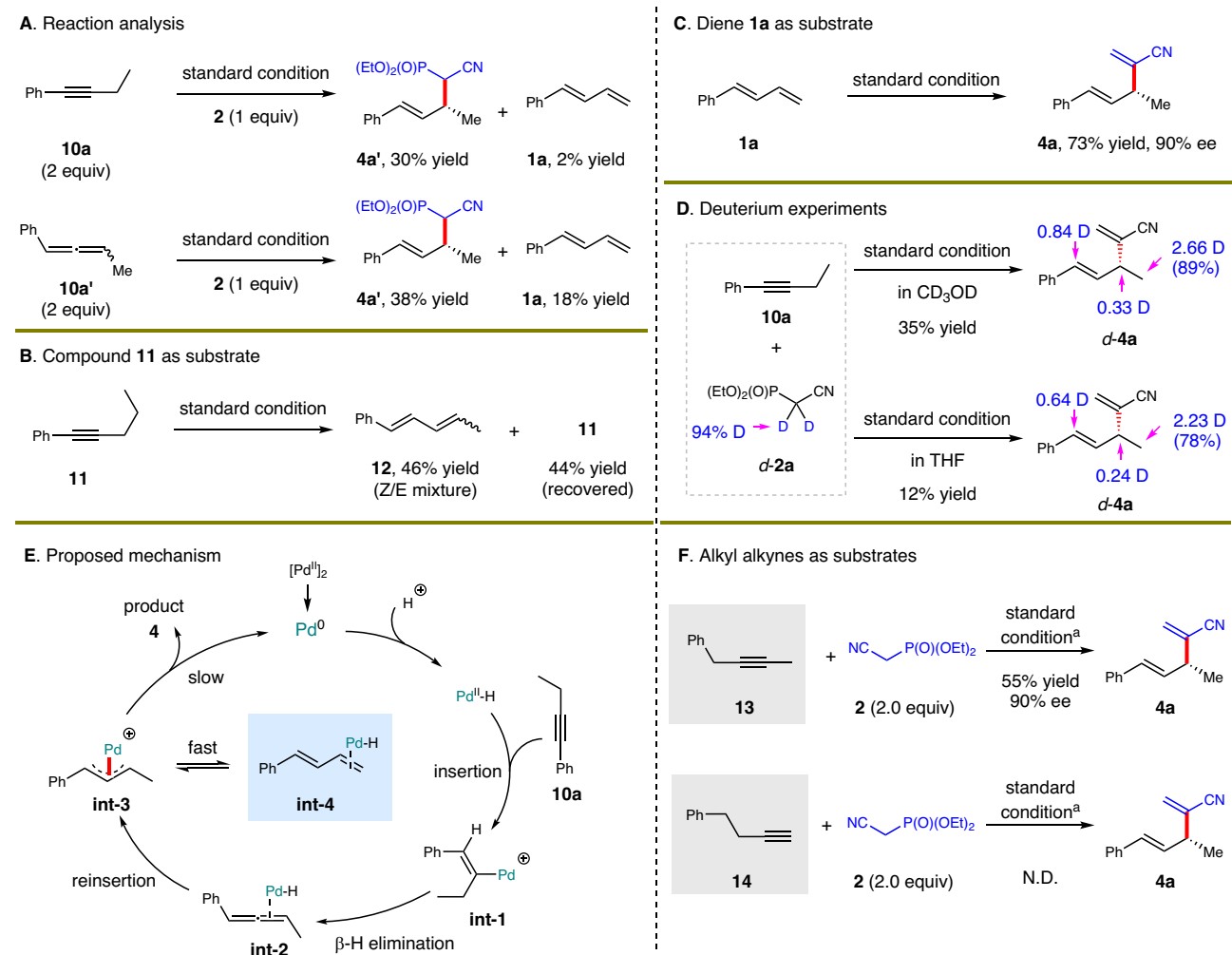

**Fig. 6 | Preliminary mechanistic studies. A** Reaction analysis. **B** Compound **11** as substrate. **C** Diene **1a** as substrate. **D** Deuterium experiments. **E** Proposed mechanism. **F** Alkyl alkynes as substrates. The standard condition was based on that shown in Fig. 5. [a][Pd(allyl)Cl]₂ (5 mol%), **L11** (10 mol%) and NaBAr$^F_4$ (10 mol%) were used.

## Methods

### General procedure for formal hydroalkenylation of 1,3-dienes

In a $N_2$-filled glovebox, [Pd(allyl)Cl]₂ (0.9 mg, 0.0025 mmol), **L11** (2.8 mg, 0.0050 mmol), NaBAr$^F_4$ (4.4 mg, 0.0050 mmol,), Ph₂P(O)OH (4.4 mg, 0.020 mmol) and DCM (0.050 mL) were added sequentially to a 4 mL vial. The resulting yellow solution was allowed to stir at ambient temperature for 1 min. Then diene **1** (0.10 mmol), diethyl (cyanomethyl)phosphonate **2** (35 mg, 0.20 mmol) and Et₃N (42 μL, 0.30 mmol) were added sequentially to the reaction. The reaction mixture continued to stir at room temperature for 48 h. After this time, aqueous formaldehyde **3a** (0.10 mL, 37% in water, 1.2 mmol) and Et₃N (14 μL, 0.10 mmol) were added to the reaction and the resulting mixture continued to stir for another 3 h at room temperature. After this time, the reaction was extracted with CH₂Cl₂ (2 mL × 3), dried over Na₂SO₄, filtered, concentrated and purified by flash silica gel chromatography to give the pure product **4**.

### General procedure for formal hydroallenylation of 1,3-dienes

In a $N_2$-filled glovebox, [Pd(allyl)Cl]₂ (0.9 mg, 0.0025 mmol), **L11** (2.8 mg, 0.0050 mmol), NaBAr$^F_4$ (4.4 mg, 0.0050 mmol), (cyanomethyl)triphenylphosphonium chloride **5** (34 mg, 0.10 mmol) and DCM/Et₃N (v/v = 1:1, 0.15 mL) were added sequentially to a 4 mL vial. The resulting yellow solution was allowed to stir at room temperature for 1 min. Then, diene **1** (0.10 mmol) was added to the reaction.

The resulting solution continued to stir at 50 °C for 36 h. After this time, the reaction solvent was removed. Next, dry THF (1.0 mL) and Et₃N (42 μL, 0.30 mmol, 3.0 equiv) were added to the reaction sequentially. The resulting solution was allowed to stir at room temperature for 3 min. Then acetyl chloride (0.30 mmol) was added to the solution slowly, and the resulting mixture continued to stir for additional 8 h at room temperature. After this time, the reaction was quenched with saturated NaHCO₃ aqueous solution (3 mL), extracted with CH₂Cl₂ (5.0 mL × 3), dried over Na₂SO₄, filtered, concentrated and purified by flash silica gel chromatography to give the pure product **7**.

### General procedure for formal hydroketenimination of 1,3-dienes

In a $N_2$-filled glovebox, [Pd(allyl)Cl]₂ (0.9 mg, 0.0025 mmol), **L11** (2.8 mg, 0.0050 mmol), NaBAr$^F_4$ (4.4 mg, 0.005 mmol), (cyanomethyl)triphenylphosphonium chloride **5** (34 mg, 0.10 mmol) and DCM/Et₃N (v/v = 1:1, 0.15 mL) were added sequentially to a 4 mL vial. The resulting yellow solution was allowed to stir at room temperature for 1 min. Then diene **1** (0.10 mmol) was added to the reaction and the resulting mixture was allowed to stir at 50 °C for 36 h. After this time, the reaction solvent was removed. Then dry THF (0.50 mL) and isocyanate **8** (0.10 mmol) were added to the reaction sequentially. The resulting reaction solution continued to stir for another 12 h at room temperature. After this time, the reaction was concentrated and purified by flash silica gel chromatography to give the pure **9**.

**A**. Convergent synthesis

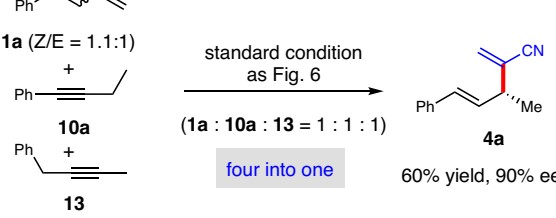

**B**. Stereodivergent synthesis

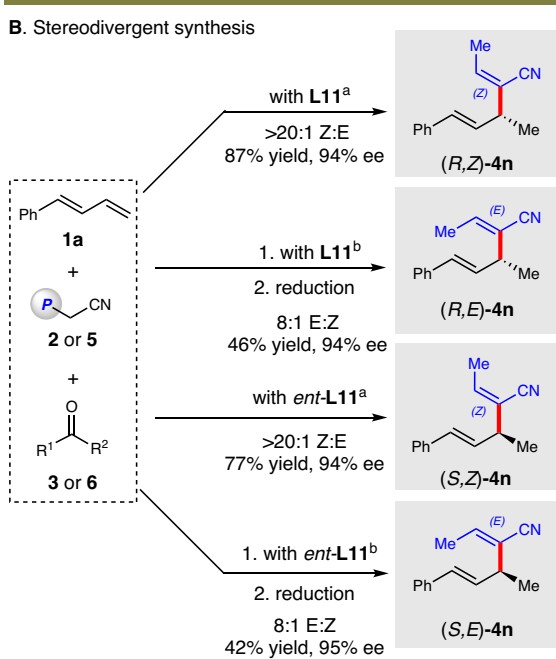

**Fig. 7 | Convergent and stereodivergent synthesis. A** Convergent Synthesis. **B** Stereodivergent synthesis. [a]Under standard condition in Fig. 2. [b]The first step was carried out to prepare **7a**. The conditions for second step: Pd₂(dba)₃ (2.5 mol%), DavePhos (5 mol%), HCO₂H (1 equiv) in *o*-Xylene at 40 °C.

## Data availability

All other data are available from the corresponding author upon request. For experimental details and procedures, spectra for all unknown compounds, see supplementary files. The X-ray crystallographic data for **4t** (CCDC 2251290), have been deposited at the Cambridge Crystallographic Data Center. These data can be obtained free of charge from The Cambridge Crystallographic Data Center via www.ccdc.cam.ac.uk/data_request/cif.

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

## Acknowledgements

We acknowledge the Shanghai Municipal Committee of Science and Technology (22ZR1475200), Natural Science Foundation of Ningbo (2023J036), National Natural Science Foundation of China (22071262, 22371292), CAS Key Laboratory of Synthetic Chemistry of Natural Substances, and Shanghai Institute of Organic Chemistry for financial support.

## Author contributions

M.-Q.T. and Z.-J.Y. contributed equally to this work. M.-Q.T. and Z.-J.Y. performed the experiments, collected and analyzed the data. Z.-T.H. conceived and directed the project. Z.-T.H. wrote the manuscript with the feedback from all authors.

## Competing interests

The authors declare no competing interests.
