## [Peer Review File · Nature Communications]

REVIEWER COMMENTS

Reviewer #1 (Remarks to the Author):

This work by He et al describes a cascade process to realize formal hydroalkenylation, hydroallylations and hydroketeniminations. Based on this design for seldomly studied hydrofunctionalizations with sp² carbon nucleophiles, a series of challenging chiral motifs were prepared smoothly, including di-, tri- and tetra-substituted alkenes, di-, tri- and tetra-substituted allenes and tri-substituted ketenimines in allyl skeletons. Generally, high reaction yields, enantioselectivities or diastereoselectivities could be observed. The authors also provided a stereodivergent synthetic protocol to prepare all four allyl motifs bearing a stereocenter and geometry-controllable olefin. For the mechanism, this work revealed an interesting mechanistic feature for the hydrofunctionalization of alkynes. That is, the alkyne would be converted into conjugated diene intermediate prior to final allylation, different from typical viewpoint that only allene intermediate was involved in the hydrofunctionalization of alkynes. Based on these facts, I support the publication of this work on Nature Communications after addressing some issues as below.

1. When the reaction was conducted with phosphonate 2, 20% diphenylphosphinic acid was adopted as additives. However, diphenylphosphinic acid was not used while Wittig reagent 5 served as the substrate. What is the reason? How about other acid additives?
2. I wondered whether Wittig reagent 5 also worked well instead of phosphonate 2 in the formal hydroalkenylation of 1,3-dienes.
3. For the product 4t in Scheme 2, the corresponding dr is only 16:1, different from that of other product like 4q, 4r and 4s, which all shared the same intermediate and should have same >20:1 dr. Please doubly check the original dr value of 4t.
4. The authors claimed that the alkyne substrate would be transformed into diene intermediate before allylic substitution. I suggest that they could conduct the reaction under standard condition without the use of nucleophiles to see whether the diene was formed. If diene intermediate was observed, it might also be a strong proof to support the present proposal.
5. Page 2, left column, "hydro functionalization" should be "hydrofunctionalization". The supplementary information was well prepared

Reviewer #2 (Remarks to the Author):

It is well known that sp^2 carbon motifs are usually not considered as suitable nucleophiles in asymmetric π -allylic substitution reactions. Thus, the enantioselective introduction of such sp^2 carbon groups to allyl skeletons via allylic substitutions or hydrofunctionalizations are very challenging, and related studies are scarce. He and coworkers report an interesting strategy to solve this issue through a sequence of hydrocarbonation and Wittig reaction to realize formal hydroalkenylation, hydroallylation and hydroketenimination of both 1,3-dienes and alkynes. With this design, di-, tri- and tetra-substituted alkenes and allenes, and tri-substituted ketenimines can be introduced to allyl skeletons in high yields and stereoselectivities, which demonstrates the value and synthetic potential of the present methods. In addition, they revealed an intriguing mechanistic aspect that alkyne undergoes present hydrocarbonation via the formation of conjugated diene intermediate instead of allene intermediate. Considering the current interests in metal hydride mediated alkene functionalization and the strength of this method, publication of this work in Nature Communications is highly recommended, pending the following minor issues are properly addressed.

1. Did the authors try to isolate and characterize the hydrocarbonation product from the first step? Is it diastereoselective?
2. One alkyl-substituted diene substrate was described and moderate reactivity with good stereoselectivity was observed. They might want to comment on the results of alkyl-substituted alkynes as electrophiles.
3. The authors used different Wittig reagents to introduce alkene and allene motifs. What is the result if Wittig reagent 5 was used instead of 2 for the construction of allyl alkenes?
4. The supplementary information is well-prepared. In addition to the yields, the actual masses of prepared products should be included.

Reviewer #3 (Remarks to the Author):

In this work, He and coworkers reported a palladium-catalyzed enantioselective sequential hydroalkylation/Wittig reaction. While palladium-catalyzed asymmetric hydrofunctionalization of unsaturated hydrocarbons with various sp^3 -pronucleophiles has been well developed, this work extends this transformation to a formal sp^2 -carbon nucleophiles. Both alkynes and conjugated dienes are suitable allyl donors for this work. A series of enantioenriched 1,4-dienes and analogues could be obtained in good yields with excellent enantioselectivities and diastereoselectivities (mostly > 90% ee, > 20:1 dr). Moreover, mechanistic studies showed that conjugated diene is one of the critical intermediates while internal alkynes are used for this transformation. This manuscript is well

organized and written, and the supporting information is well done. Therefore, I recommend acceptance of this manuscript in Nature Communications after minor revision.

1. The authors have shown the substrate scope to the preparation of di-, tri- and tetra-substituted alkenes in Scheme 2. What about the result if an unsymmetric ketone (such as acetophenone) was used?
2. It is quite interesting that both E and Z-isomers of conjugated diene 1 gave the products 4 with the identical absolute configuration. Further discussion on this result was encouraged. Did the authors observe Z to E isomerization for the olefin units of compound 1 during the reaction process?
3. As shown in Scheme 6A, while internal alkyne 10a was used, conjugated diene 1a could be detected. If allene intermediate was treated with standard conditions, could conjugated diene 1a be observed?
4. The content for the bottom-right corner of Scheme 1b is misleading to some extent. According to the results shown in this work, conjugated diene should be one of the intermediates for the hydroalkenylation of alkynes. Since the conjugated diene should be generated from allene intermediate, it's not suitable for the authors to give a conclusion of "not allene, typical mechanistic viewpoint", as shown in Scheme 1b.
5. For Scheme 6A, it should be "equiv" instead of "euqiv".

Point-by-point responses to the reviewers

Reviewer 1

1. The reviewer wondered why 20% diphenylphosphinic acid was adopted with phosphonate **2** but not with Wittig reagent **5** and if other acid additives were tested. We think the major reason is that comparing with phosphonate **2**, Wittig reagent **5** contains a more acidic C-H bond and can easily generate critical PdH catalyst. In this case, acidic condition might be unfavorable for the formation of nucleophilic anion. But for phosphonate **2**, additional diphenylphosphinic acid is required to promote the generation of PdH. We did test other acid additives for the reaction with Wittig reagent **5** (as shown below), but no better results were observed.

2. The reviewer asked the effect of the use of Wittig reagent **5** instead of phosphonate **2** for the formal hydroalkenylation of 1,3-diene. We conducted this test and product **4a** was obtained in a slightly decreased yield but with the same high enantioselectivity.

3. The reviewer said that for the product **4t** in Scheme 2, the corresponding dr is only 16:1, different from that of other product like **4q**, **4r** and **4s**, which all shared the same intermediate and should have same >20:1 dr. We have doubly checked the dr of **4t** and it is indeed 16:1. The reason is that the α -stereocenter of sugar-derived aldehyde substrate would slightly racemize under the basic condition.
4. The reviewer suggested to conduct a reaction under standard condition without the use of nucleophiles to see whether the diene was formed, which might be used as an additional proof to support the present mechanistic proposal. We have carried out this experiment as shown below, but failed to observe diene intermediate. We guess the nucleophile **2** might also plays an important role in the generation of PdH catalyst. Thus, the absence of **2** would affect the transformation.

5. The reviewer pointed out that the word “ hydro functionalization ” should be “hydrofunctionalization”. We have revised it.

Reviewer 2

1. The reviewer wondered if we isolated and characterized the hydrocarbonation product from the first step and if it was diastereoselective. We did fully characterize this intermediate and the dr was about 1:1, due to the strong acidity of α C-H bond that could undergo facile racemization. The related data of this compound **4a'** has been provided in Supplementary Information 8.1.

2. The reviewer asked to comment on the results of alkyl-substituted alkynes as electrophiles. We have described two cases on the use of alkyl-substituted alkynes as electrophiles as shown below. When alkyne bearing a benzyl group was used, the reaction proceeded well and product **4a** was formed in 55% yield and 90% ee. When a terminal alkyne bearing a phenylethyl substituent was used, however, no product was observed. These facts suggest that the existence of vicinal aryl unit to alkyne is important, due to the easy formation of aryl-conjugated diene intermediate. Or the transformation will become challenging.

These results and related comments have been added to the main text, and the description is shown as below:

“We did prepare **4a** from **13** in a moderate yield and high enantioselectivity (Fig. 6F). However, when terminal alkyl alkyne **14** was adopted as the electrophile, no product was detected. These facts suggested that the existence of vicinal aryl unit to alkyne is important, due to the easy formation of aryl-conjugated diene intermediate.”

3. The reviewer asked the result when Wittig reagent **5** was used instead of **2** for the construction of allyl alkenes. We conducted this test and product **4a** was obtained in a slightly decreased yield but with the same enantioselectivity.

4. The reviewer suggested to add the actual masses of prepared products to the Supplementary Information. We have added them as suggested.

Reviewer 3

1. The reviewer asked to test an unsymmetric ketone, such as acetophenone for the preparation of compound **4** in Scheme 2. We have conducted this reaction with acetophenone as shown below, but did not observe any product, presumably due to the increased steric hindrance. At present, only special cyclic ketones can work well with the method.

2. The reviewer wondered if we observed Z to E isomerization for the olefin units of compound **1** during the reaction process. We have prepared (*Z*)-**1a** and tested the potential isomerization as shown below. When then reaction was conducted with (*Z*)-**1a** as the substrate but without nucleophile **2**, no (*E*)-**1a** was detected and (*Z*)-**1a** was recovered in 89% yield. Considering that nucleophile **2** might facilitate the generation of PdH catalyst, the other experiment with both (*Z*)-**1a** and nucleophile **2** as the substrates under standard condition was conducted. Then 26% of (*E*)-**1a** was observed along with hydroalkylation product **4a'** in 62% yield and (*Z*)-**1a** recovered in 8% yield. This fact suggests that (*Z*)-**1a** might undergo the isomerization to (*E*)-**1a** during the reaction.

These experiments have been added to the Supplementary Information and related comments have been added to the main text and are described as below:

“As the geometry of internal olefin in diene substrate **1** did not affect the reaction, presumably due to the facile isomerization of (Z)-**1** into (E)-**1** (see Supplementary Information 8.6 for details), Z/E mixtures of **1** were directly used as substrates for present transformation.”

3. The reviewer asked if allene intermediate was treated with standard conditions, could conjugated diene **1a** be observed. We adopted allene intermediate **10a'** as the substrate and tested the transformation under the same condition described in Scheme 6A. Just as the reviewer thought, 18% of diene **1a** was detected, along with 38% hydroalkylation intermediate **4a'** and 66% **10a'** recovered as shown below. This experiment is also a proof to support the formation of 1,3-diene in the reaction, and the result has been added to the main text.

The newly added description is shown as:

“However, when we analyzed the results of hydroallenylation reaction of alkyne **10a**, trace amount of conjugated diene **1a** was observed (Fig. 6A). With allene intermediate **10a'** as the substrate for hydrofunctionalization, diene **1a** was obtained in 18% yield. These facts led us to hypothesize that diene might be a critical intermediate in the transformation of alkyne.”

4. The reviewer thought the content for the bottom-right corner of Scheme 1b was misleading to some extent, because conjugated diene should be one of the intermediates for the hydroalkenylation of alkynes and was generated from allene intermediate. Agreeing with the reviewer, we have revised the original description to clear out possible misleading. The revised figure is now showing as below:

5. The reviewer pointed out that it should be “equiv” instead of “euqiv” in Scheme 6A. We have revised this typo.

REVIEWERS' COMMENTS

Reviewer #1 (Remarks to the Author):

The authors have solved all the issues provided in the first-round review and now I believe the manuscript is suitable for publication.

Reviewer #2 (Remarks to the Author):

The issues raised by the reviewers are properly resolved. This paper is ready to publish.

Reviewer #3 (Remarks to the Author):

The authors have addressed all the questions arised by the reviewers. It was suggested to be accepted for publication now.